# The Effectiveness of Therapeutic Vaccines for the Treatment of Cervical Intraepithelial Neoplasia 3: A Systematic Review and Meta-Analysis

**DOI:** 10.3390/vaccines10091560

**Published:** 2022-09-19

**Authors:** Cathy Ventura, Ângelo Luís, Christiane P. Soares, Aldo Venuti, Francesca Paolini, Luísa Pereira, Ângela Sousa

**Affiliations:** 1CICS-UBI–Health Science Research Centre, University of Beira Interior, Av. Infante D. Henrique, 6200-506 Covilhã, Portugal; 2Grupo de Revisões Sistemáticas (GRUBI), Faculdade de Ciências da Saúde, Universidade da Beira Interior, 6200-506 Covilhã, Portugal; 3Department of Clinical Analysis, School of Pharmaceutical Sciences, São Paulo State University (UNESP), Campus Ville, Araraquara 14800-903, SP, Brazil; 4HPV-UNIT-UOSD Tumor Immunology and Immunotherapy, IRCCS Regina Elena National Cancer Institute, 00144 Rome, Italy; 5CMA-UBI-Centro de Matemática e Aplicações, Universidade da Beira Interior, 6200-001 Covilhã, Portugal; 6C4-UBI, Cloud Computing Competence Centre, University of Beira Interior, 6200-284 Covilhã, Portugal

**Keywords:** cervical cancer, clinical trials, meta-analysis, systematic review, therapeutics vaccines

## Abstract

Cervical cancer (CC) is a disease that affects many women worldwide, especially in low-income countries. The human papilloma virus (HPV) is the main causative agent of this disease, with the E6 and E7 oncoproteins being responsible for the development and maintenance of transformed status. In addition, HPV is also responsible for the appearance of cervical intraepithelial neoplasia (CIN), a pre-neoplastic condition burdened by very high costs for its screening and therapy. So far, only prophylactic vaccines have been approved by regulatory agencies as a means of CC prevention. However, these vaccines cannot treat HPV-positive women. A search was conducted in several databases (PubMed, Scopus, Web of Science, and ClinicalTrials.gov) to systematically identify clinical trials involving therapeutic vaccines against CIN 3. Histopathological regression data, immunological parameters, safety, DNA clearance, and vaccine efficacy were considered from each selected study, and from the 102 articles found, 8 were selected based on the defined inclusion criteria. Histopathological regression from CIN 3 to CIN < 1 was 22.1% (95% CI: 0.627–0.967; *p*-value = 0.024), showing a vaccine efficacy of 23.6% (95% CI; 0.666–0.876; *p*-value < 0.001). DNA clearance was assessed, and the risk of persistent HPV DNA was 23.2% (95% CI: 0.667–0.885; *p*-value < 0.001). Regarding immunological parameters, immune responses by specific T-HPV cells were more likely in vaccinated women (95% CI: 1.245–9.162; *p*-value = 0.017). In short, these studies favored the vaccine group over the placebo group. This work indicated that therapeutic vaccines are efficient in the treatment of CIN 3, even after accounting for publication bias.

## 1. Introduction

Cancer is a devastating disease, being associated with a burden on affected individuals and their families as well as health care systems [1]. The incidence of cancer-related to viral infections accounts for 15–20% of cases globally, where cervical cancer (CC) and hepatocellular carcinoma represent 80% of virus-linked cancers [2]. Cervical cancer is the fourth most common cancer of women worldwide and is mostly caused by the human papillomavirus (HPV), which is highly transmissible through unprotected sexual contact [3]. In 2020, 17.6% of new cancer cases in women were CC, with 10.7% of associated deaths, being considered a public health global problem [4]. In addition, cervical intraepithelial neoplasia (CIN) is a pre-neoplastic condition linked to HPV infection with an incidence rate higher than that of CC and with high costs for health organizations [5,6].

HPV belongs to the *Papillomaviridae* family, a group of small, non-enveloped, double-stranded DNA viruses [1]. The genetic material responsible for the expression of HPV E1, E2, E3, E4, E5, E6, and E7 proteins, which are involved in viral replication, the regulation of transcription, and oncogenesis [7], are encapsulated by an icosahedral coat composed of structural proteins, L1, and L2. HPVs exhibit tissue specificity and infect the skin epithelium and mucosa. More than 400 types of HPV have been identified, with at least 14 high-risk (HR) types that can cause cancer [8,9]: HPV 16, 18, 31, 33, 34, 35, 39, 45, 51, 52, 56, 58, 59, 66, 68, and 70 [10]. Of all these types, HPV-16 and 18 are strongly associated with cancer progression [11], accounting for 79 to 100% of cases of HPV-associated CC [3].

After the virus entrance in the epithelial cells, the viral DNA starts the replication, and a small amount of E1, E2, E6, and E7 proteins is produced before the infection even happens. Then, E2 invokes E1 to promote an increase in the number of viral episome copies [12,13]. E6 and E7 oncoproteins are key factors that trigger the malignant phenotype of HPV-positive cells [14]. E6 is considered the critical element in the regulation of the viral life cycle and unleashes the process of HPV tumorigenesis through the degradation of the tumor suppressor p53. On the other hand, E7 allows HPV to escape the G1-S checkpoint by inactivating the retinoblastoma tumor suppressor protein (pRB1) [15,16]. The viral life cycle is ended when the L1 and L2 proteins are expressed in the uppermost layer of the epithelium. Thus, the viral genome is encapsulated, and virions are released in the surface layers through cellular desquamation [17,18].

When a HPV infection becomes persistent, a progression to pre-malignant glandular or squamous intraepithelial lesions can emerge. From a histopathological perspective, these lesions can be classified as cervical intraepithelial neoplasia (CIN), which is subdivided into CIN 1—mild dysplasia; CIN 2—moderate to severe dysplasia; CIN 3—severe dysplasia and carcinoma in situ, and can progress to cancer stage [19]. Despite the likelihood of the increased progression of lesions, a proportion can still regress. An infection with HPV is a risk factor for persistent and/or progressive cervical dysplasia, and it has been proposed that the integration of HPV DNA with host DNA is a key factor for cervical carcinogenesis [8].

There are three marketed prophylactic vaccines (i) Cervarix (GlaxoSmithKline, Middlesex, UK), a bivalent vaccine against HPV 16 and 18; (ii) Gardasil (Merck Inc., Rahway, NJ, USA), a quadrivalent vaccine against HPV 6, 11, 16, and 18, and (iii) Gardasil 9 (Merck Inc.), a nonavalent vaccine against HPV 6, 11, 16, 18, 31, 33, 45, 52, and 58 [20]. The principle of these preventive vaccines focuses on the delivery of virus-like particles based on the recombinant L1 protein to stimulate neutralizing antibodies against HPV and develop immune memory [21]. However, prophylactic vaccines mostly activate the humoral pathway of the immune system and are not effective or present a therapeutic effect against an ongoing HPV infection [3]. In addition, the existing CC therapies, such as chemotherapy and radiotherapy, are invasive or present toxicity in healthy cells and consequently side effects for the patients, pointing to the need for new, safe, and more efficient approaches.

Prophylactic vaccines are generally given to healthy individuals and have little potential when the disease is already present. Therapeutic cancer vaccines, however, are given to cancer patients and are designed to eradicate cancer cells. This technology can induce immune responses to HPV proteins with the aim of removing HPV infection by killing HPV + cells in a lesion or tumor [22,23]. E6 and E7 oncoproteins are good targets for CC immunotherapy as they are primordial to the onset and progression of malignancy. Several strategies for HPV therapeutic vaccines designed to enhance CD4 + and CD8 + T-cell responses have been investigated, including genetic (i.e., DNA/RNA/virus/bacterial), and protein-, peptide- or dendritic cell-based vaccines. It is conceivable that these vaccines would have their greatest efficacy when applied to pre-neoplastic lesions where tumor-induced immunosuppression is less effective. It is important to note that the standard treatment of CIN 2/3 is surgical excision that is associated with considerable morbidities, i.e., low birth weight, pre-term births, increased deaths, and more caesarian section [24]. However, none of the proposed vaccines in clinical trials have yet been approved [25].

This systematic review with a meta-analysis of randomized controlled trials (RCTs) aims to evaluate the efficacy of therapeutic vaccines against CIN 3.

## 2. Materials and Methods

### 2.1. Search Strategy

This systematic review was conducted according to the Preferred Reporting Items for Systematic Reviews and Meta-Analyses (PRISMA) statement [26]. The search was managed until 28 March 2022 in the following databases: PubMed, Scopus, Web of Science, and ClinicalTrials.gov. For each, this sequence of keywords was applied: (therapeutic AND vaccine* AND (HPV OR (cervical cancer)) AND CIN 3) selecting the option (Randomized Clinical Trials (RCT)) in all databases.

### 2.2. Study Selection, Inclusion and Exclusion Criteria and Data Extration

Two reviewers independently reviewed the selected studies for inclusion eligibility using standardized data collection forms. The following data were retrieved from each study: (i) author, (ii) year of publication, (iii) number of study participants, (iv) type of the vaccine and placebo administered, (v) study duration/location (vi) CIN, (vii) main outcomes (histopathological regression, immunological parameters, safety, DNA clearance and efficacy). In the face of disagreement between the authors, a third author was called to discuss and find a consensus to include or exclude the study in question.

To validate the articles, they had to meet the following eligibility criteria: (i) RCT, (ii) include participants with CIN 3, and (iii) include the assessment of CIN 3 regression in both control and experimental groups. Titles and abstracts that did not fit the criteria were eliminated without further review.

The main outcomes of this meta-analysis were clinical efficacy based on the lesion regression, viral load reduction, and immunogenicity, in particular HPV-T cell response.

### 2.3. Risk of Bias Assessment

We assessed internal validity using Cochrane Collaboration’s Risk of Bias Tool. This tool classifies the risk of bias in RCTs included in meta-analysis as high risk, unclear, or low risk in seven domains: random sequence generation, allocation concealment, blinding of participants and personnel, blinding of outcome assessment, incomplete outcome, data selective reporting, and other sources of bias [27]. This classification was independently assigned by two authors, and discrepancies were resolved through discussions between the authors or through consultation with a third researcher. The results of the risk of bias assessment are presented in a risk of bias summary and in a risk of bias graph that was produced using the software Review Manager 5.3 (Version 5.4.1) (London, UK) (Cochrane Training, https://training.cochrane.org/online-learning/core-software-cochrane-reviews/revman/revman-5-download/download-and-installation) (accessed on 15 September 2022).

### 2.4. Statistical Analyses

Data statistical analyses were performed using Comprehensive Meta-Analysis, version 2.0 (https://www.metaanalysis.com/) (accessed on 31 January 2022). Forest plots were generated to illustrate the study-specific effect sizes (risk ratio—RR or odds ratio—OR) along with a 95% confidence interval (CI). In general, the random-effects model was used; however, when a fixed-effects model would be more useful, i.e., when there was little variance in effect sizes, the software automatically converted the random-effects into a fixed-effects model [28]. Heterogeneity between trial results was tested using the *Q*-statistic (also referred to as Cochrane’s *Q*). Under the hypothesis that homogeneity exists (null hypothesis), *Q* will follow a χ^2^ distribution. A *p*-value of less than 0.05 leads to the rejection of the null hypothesis and therefore indicates that some (undetermined) degree of heterogeneity exists. Higgins et al. [29] I^2^ was used as a measure of inconsistency across the findings of the included studies. I^2^ has a range of 0 to 100%, and values of 25%, 50%, and 75% are considered to indicate low, moderate, and high heterogeneity, respectively [29]. I^2^ reflects the proportion of observed dispersion which was due to the heterogeneity.

Three analyses were performed to assess the potential impact of publication bias on the meta-analysis. One analysis was a funnel plot in which the log of RR or OR was plotted against the corresponding standard error (SE). In the absence of publication bias, the studies would have been symmetrically distributed about the pooled effect size [30,31]. Since the interpretation is largely subjective, the Egger’s regression test was also performed [32]. Finally, the trim and fill approach was applied, which uses an iterative procedure to remove the most extreme small studies from the positive or negative side of the funnel plot and recalculate the effect size at each iteration until the funnel plot is symmetrical in terms of the new effect size, yielding an unbiased estimate of the pooled effect size. This approach allows for obtaining the best estimate of the unbiased pooled effect size and lends itself to an intuitive visual display as it creates a funnel plot that includes both the observed studies (shown as blue circles) and the imputed studies (shown as red circles). As a result, this visual display shows how the pooled effect size shifted when the imputed studies were included [33,34].

The sensitivity analysis was also performed by removing each study one at a time to evaluate the stability of the results.

## 3. Results

### 3.1. Study Identification and Selection

Our search retrieved 102 records, of which 37 were duplicates and were then excluded. The titles and abstracts of the remaining 65 published articles were screened, and 8 articles were found to meet the inclusion criteria and were assessed for eligibility via full-text evaluation (Figure 1). Another 49 articles were excluded through the title and abstract, and, after that first selection, 8 records in total did not meet the inclusion criteria after this second full-text review. Data from the remaining 8 studies were included in the primary analysis.

### 3.2. Included Studies and Characteristics

The selected studies focused on vaccines developed for the regression of CIN 3 (Table 1), in which the year of publication, the name of the vaccine, the type of vaccine, the number of participants in both the control and experimental groups, and the CIN of the volunteers involved were collected. The included studies fall between the years 2004 and 2021, during which time, 789 patients were involved in RCTs. In these RCTs, the primary outcomes involved the analysis of histopathological regression of patients diagnosed with CIN 1+, 2/3, and 3 who took the study vaccine compared with a placebo.

### 3.3. Risk of Bias Assessment

The risk of bias of each included RCT was assessed with the Cochrane Library tool. The results found in the assessment of the risk of publication bias from the included studies are summarized in Figure 2 and Figure 3. In general, all the trials satisfied the seven domains of bias defined by Cochrane Collaboration. All the included trials claimed to be randomized, but only five trials detailed the randomization process and so were classified as low risk in the random sequence generation domain. Other potential sources of bias were found such as financial support. It is important to note that the assessment of publication bias is a subjective task since it is based on the personal judgments of the review authors.

### 3.4. Outcomes

The effect sizes were summarized as RR or OR and associated with 95% CI. A RR less than one or an OR great than one suggested vaccine protection against a clinical endpoint. Consistent with definitions used in vaccine RCTs, efficacy was estimated as (1-RR) and expressed as a percentage.

#### 3.4.1. Clinical Efficacy Was Evaluated According to the Histopathological Regression to CIN ≤ 1, and Vaccine Efficacy for Complete Resolution

The histological complete resolution or partial response (CIN 1) was obtained by measuring the lesion before and after vaccination in women who were confirmed to have CIN 2 or 3 at baseline. After this regression, the vaccine efficacy (VE) for complete resolution was calculated.

Eight studies assessed the histopathological regression to CIN ≤ 1 and VE for complete resolution.

Vaccines showed a statistically significant reduction of 22.1% in CIN 2/3 incidence compared with placebo (RR: 0.779; 95% CI: 0.627–0.967; *p*-value = 0.024) (Figure 4). However, there was statistically significant heterogeneity among included trials (*Q* = 16.31; I^2^ = 63.21%; τ^2^ = 0.047; *p*-value = 0.012).

The pooled RR for confirmed cases of CIN 1/2/3 in the vaccinated group compared with the unvaccinated group was 0.764 (95% CI; 0.666–0.876; *p*-value < 0.001) (Figure 5), corresponding to a pooled efficacy of 23.6% for complete resolution and indicating a statistically significant benefit with vaccine use. There was a non-significant heterogeneity (*Q* = 5.818; *p*-value = 0.444; I^2^ < 0.01%, τ^2^ < 0.01).

#### 3.4.2. DNA Clearance

Viral load was measured using PCR amplification to assess the clearance of HPV DNA from the cervical biopsy after vaccination. Four studies reported on DNA clearance. Viral DNA clearance of CIN 2/3 was significantly greater in the vaccine-treated group than in the placebo group. In addition, Figure 6 shows that the pooled RR for the risk of persistent HPV infection (detection of some HPV DNA) was 0.768 (95% CI: 0.667–0.885; *p*-value < 0.001) equivalent to a 23.2% lower risk of persistent HPV DNA for vaccine recipients compared with control recipients. There was a non-significant heterogeneity (*Q* = 1.273; I^2^ < 0.01%; τ^2^ < 0.01; *p*-value = 0.736).

#### 3.4.3. Immunogenicity

Immunogenicity is one of the key features of therapeutic vaccines as it represents the potential of the vaccine to induce virus-specific T cell response, in particular HPV-T cell immune response.

Vaccinated women were more likely to develop HPV-T cell immune response than were unvaccinated women (OR: 3.381; 95% CI: 1.245–9.162; *p*-value = 0.017) (Figure 7). There was a non-significant heterogeneity (*Q* = 2.499; I^2^ < 0.01%; τ^2^ < 0.01; *p*-value = 0).

#### 3.4.4. Safety

Overall, all vaccines were well tolerated without vaccine-related serious adverse events. No serious adverse events (Grade 3/4) related to the vaccination were reported. No dose-limiting toxicities were observed.

### 3.5. Sensitivity Analysis

The sensitivity analysis was performed by excluding one or more studies from the analysis to see how this affected the results. Special attention was also given to studies identified as outliers. The sensitivity analysis showed that the pooled effects of the effectiveness of DNA therapeutic vaccines endpoints, efficacy for complete resolution, histopathological regression to CIN ≤ 1, and DNA clearance, did not change substantially if a single or a few studies were omitted (Figure 8). Regarding HPV-T cell response after vaccination, the conclusion reached by excluding the study of Y. Ikeda et al. [39] (Table 2) was more favorable to the vaccination (OR: 7.835; 95% CI: 1.632–37.61; *p*-value = 0.01) when compared with the conclusions obtained for the global analysis (pooled OR: 3.381; 95%CI: 1.245–9.182; *p*-value = 0.017).

Overall, the sensitivity analysis demonstrated that the findings of the meta-analysis of the effectiveness of DNA therapeutic vaccines for cervical intraepithelial neoplasia 2/3 on the defined outcomes of this work are robust.

### 3.6. Publication Bias

To analyze the publication bias, funnel plots were generated for the outcomes (Figure 9) considering the trim and fill adjustment. It was observed that for all the outcomes, there were not the same number of studies on the right and on the left, so for each one of these measures, some studies were imputed on the left or on the right to “adjust” the funnel plots for the absence of publication bias. Therefore, by observing the funnel plots, it was possible to verify that publication bias could not be completely excluded. The adjusted values of RR and OR are presented in Table 2. In addition to the visual inspection of the funnel plots, the presence of publication bias was explored using Egger’s regression test. For each effect size, Egger’s test for a regression intercept gave a *p*-value > 0.05, indicating no evidence of publication bias (Table 3).

## 4. Discussion

Primary and secondary strategies for preventing CC remain key in reducing the burden of the disease, but therapeutic vaccines could avoid invasive surgery. The initial search of several databases on clinical trials for the efficacy of therapeutic vaccines against CIN 3 resulted in 102 articles, of which only 8 were included in this meta-analysis. Several therapeutic vaccine typologies were explored in different studies, including the use of plasmid DNA encoding E6 and E7 oncoproteins, recombinant proteins, and virus-like particles (VLPs). The present work evaluated the therapeutic efficacy of these vaccine technologies by considering and comparing the following outcomes: (i) histopathological regression, (ii) VE, (iii) DNA clearance, (iv) immunogenicity, and (v) safety.

Virus-like particles are multi-protein supra-molecular structures and present varied virus characteristics that can be applied in vaccine development strategies. VLP-based vaccines exclude the virus ability to replicate, since it lacks the viral genome, making it a safe model for vaccine development [43]. Gardasil is a prophylactic vaccine produced by expressing the HPV L1 capsid protein [44]. During the initial search, several clinical trials based on this Gardasil vaccine were signaled. However, some of these studies were discharged because they evaluated the preventive capacity of this vaccine. Only two clinical assays studied the therapeutic potential of Gardasil. One of them was not included in this systematic review with meta-analysis because it did not meet the defined inclusion criteria, such as the CIN regression. The study involved 419 volunteers in the experimental group and 446 in the placebo group, all seropositive and DNA positive for HPV16 or HPV18. The obtained results indicated that the Gardasil vaccine did not affect the CIN condition in women with ongoing HPV 16/18 infection. Thus, this study concluded that use of prophylactic vaccines cannot reduce or even regress the progression of HPV-infected individuals [45]. Nevertheless, Karimi-Zarchi and coworkers observed that the Gardasil vaccine allowed partial or complete regression for CIN 1–3 in a group of 152 volunteers. The VE was 54.9% for the group with CIN 1, 63.3% for CIN 2 and 52.5% for CIN 3. However, HPV DNA clearance was not evaluated, and the possibility of CIN recurrence cannot be excluded [36]. Meanwhile, Kaufmann and colleagues studied a different vaccine, improving the VPL technology through the truncation of the C-terminal of the L1 protein fused to the N-terminal part of the E7 protein (E71–55) of HPV 16. This therapeutic strand allows a significant increase in the humoral and cellular component of the immune system. The chimeric VLP vaccine promotes regression to CIN < 1 and eliminates viral DNA from infected cells, proving to be effective against HPV-related CC with a VE of 35.90% [38].

A vaccine based on recombinant E6/E7 fusion proteins, and an adjuvant was explored by Frazer and coworkers and revealed an improved specific antibody response, and also developed E6 or E7 specific T cells [40]. However, further studies are needed to establish the histopathological regression and determine the efficacy of the vaccine. Other vaccine typology was investigated by Ikeda and collaborators. They explored the use of *Lactobacillus casei* (*L. casei*) to present specific antigens to APCs [46]. Thus, GBLBL101c vaccine was developed from a heat attenuated recombinant *L. casei* expressing mutant HPV16 E7 [39]. A plasmid encoding E7 HPV 16 protein was transferred to *L. casei* by electroporation. Subsequently, *L. casei* expressed E7 HPV 16 on its surface [46]. The attenuated *L. casei* was purified and dried into a powder and stored in a capsule designed to degrade in the gut containing 250 mg of the vaccine. GLBL101c was therefore administered orally. This vaccine allowed the development of Th1 cells, leading to a significant regression of the CIN presenting a VE of 100% [39].

DNA vaccines are considered the third generation of vaccines since they stimulate both humoral and cellular pathways of the immune system, have low production cost, are thermostable, and are easy to deliver compared with conventional vaccines [47]. These vaccines allow for the delivery of genetic information to develop an immune response against one or more specific antigens [48]. To amplify the immune response activation, the DNA vaccine can be encapsulated and condensed within delivery systems based on polymers, liposomes, live attenuated viruses, among others [49,50]. These DNA vehicles improve tissue penetration and cellular uptake due to their nanoscale and protect nucleic acids from intra- and extracellular barriers [51]. For instance, Tipapkinogen Sovacivec (TS) therapeutic HPV vaccine is based on the modified Ankara vaccine (MVA) virus being used as a genetic vaccine delivery system. TS has inserted genes encoding three proteins: human cytokine IL-2, and the modified HPV 16 proteins E6 and E7. Harper and colleagues evaluated histopathological regression, HPV DNA clearance, and VE of TS. They found a partial histopathological regression of 11.6% and complete histopathological regression of 24%. The vaccine showed superior HPV DNA clearance in the experimental group compared to control for both CIN 2/3 and 3. This clearance was maintained for 2.5 years after vaccination, and this vaccine showed a VE of 60.0% [37].

ZYC101a consists of an encoded HPV 16 E7 protein via pDNA encapsulated in polylactide co-glycolide (PLG) microparticles. Francisco Garcia and collaborators conducted a study involving 161 volunteers. Histopathological regression was higher in the ZYC101a groups compared to placebo (43% versus 27%). In addition, in women younger than 25 years, regression was higher in the experimental group (70%) compared with the control group (23%). The immune response showed a 2-fold increase of HPV 16-specific CD8 + T-cell production. Thus, the VE of this vaccine was 37.21% [41]. Matijevic and colleagues studied the same vaccine by modifying the expression for E6 HPV 16 and E7 HPV 18 protein. They observed that ZYC101a administration promoted histopathological regression of CIN 2/3 with the two doses tested 100 and 200 µg. They also verified that the immune response increased 2-fold in CD8 + T cells specific for HPV 16 [42].

Another vaccine exploiting DNA technology is the VGX-3100 (33). This vaccine consists in the administration of a final dose of 6 mg (3 mg of the plasmid expressing the E6 and E7 protein of HPV 16 and 3 mg of the plasmid expressing the E6 and E7 protein of HPV 18) through intramuscular route followed by electroporation. Involving 154 patients, this vaccine was well tolerated, manifesting common adverse effects at the administration site. The histopathological regression parameter was evaluated, and the result was 48.2% in the experimental group and 30.0% in the control group. The immunological component was improved, namely the specific T-cell response, which was increased 9.5-fold compared to baseline. The clearance of HPV DNA was also checked, showing an elimination of HPV-positive cells of 39.5% in the experimental group and only 15.40% in the placebo group. This vaccine had a VE of 37.76% [35].

Overall, the present study shows that the use of therapeutic vaccines against HPV-related CC can provide effective immune responses that lead to the histopathological regression of CINs and the elimination of HPV DNA from infected cells. The results observed with DNA vaccines can be related to the fact that some vaccines are applied without delivery systems and other make use of DNA delivery systems with different performance in the process of cellular internalization. Undoubtedly, the use of delivery systems can bring several advantages for DNA vaccines, not only to efficiently condense, protect, and carry the DNA but also to target deliver these vaccines into APCs, being considered strong adjuvants in the intended immune responses activation [52,53]. For instance, oligosaccharide-lectin interaction has already been explored for targeting specific delivery systems, and many glycoconjugates have demonstrated the potential of this “glycotargeting”. Mannose receptors are highly expressed on the surface of some cells, such as macrophages and dendritic cells, which are responsible for the activation of cellular and humoral pathways of the immune system [53]. Thus, the efficiency of pDNA vaccines can be improved by applying a mannosylated delivery system for targeting to antigen-presenting cells. Additionally, some vaccines have shown promising therapeutic results when explored in combination with immunotherapy. The GX-188E, a HPV therapeutic DNA vaccine (consisting of a tissue plasminogen activator signal sequence, an FMS-like tyrosine kinase 3 ligand, and shuffled E6 and E7 genes of HPV type [50]), plus pembrolizumab showed antitumour activity against recurrent or advanced cervical cancer [51]. This combination therapy could also represent a new potential treatment option for this patient population. The ISA101 vaccine consists of HPV-16 E6 and E7 synthetic long peptides, and although this vaccine is focused on eradicate HPV-16+ pre-malignant vulvar lesions, when combined with nivolumab may enhance adaptive antitumor immunity via expansion of HPV-specific T cells [52].

The sensitivity analysis results showed that the efficacy of therapeutic vaccines in histopathological regression for CIN < 1 did not change when removing either study, implying that the results found are robust. The risk of publication bias cannot be excluded, showing that there are factors that may influence the results of this meta-analysis. However, this systematic review with meta-analysis demonstrated that those therapeutic vaccines could efficiently reduce CIN and were excellent approaches to treating women with HPV-related CC.

Previous systematic reviews have also demonstrated the therapeutic potential of DNA vaccines against CIN 3. Indeed, therapeutic vaccines have the potential to be a novel therapy for cancer. Despite currently available treatments, such as chemotherapy or radiotherapy, the recurrence rate remains high (50% of cases) and the 5-year survival rate is low. However, one DNA vaccine, MEDI0457 (plasmid VGX-3100 associated with a plasmid expressing IL-12), gave an estimated disease progression-free survival (PFS) of 88.9% [54]. Although therapeutic vaccines have enormous potential in the treatment of CC, they are not yet licensed. Prophylactic vaccines can reduce the prevalence of a persistent HPV infection in healthy women or after CIN treatment [55,56] but cannot eliminate the CC induced by persistent HPV infection when it is already established in the patient. Thus, the development of therapeutic vaccines is the ultimate goal for curing malignant disease [57].

## 5. Conclusions

Cervical cancer remains a very prevalent disease worldwide, for which only prophylactic vaccines are available until now. This systematic review and meta-analysis provided an overview of the therapeutic vaccines under clinical assays. Among the most varied technologies evaluated and for the chosen parameters, the vaccine group was always the favored one. These results reflect the importance of investigating therapeutic vaccines against CC in order to treat HPV-positive patients, especially in the initial stages of cancer such as CIN, also reducing the use of invasive surgical procedures for their treatment.

## Figures and Tables

**Figure 1 vaccines-10-01560-f001:**
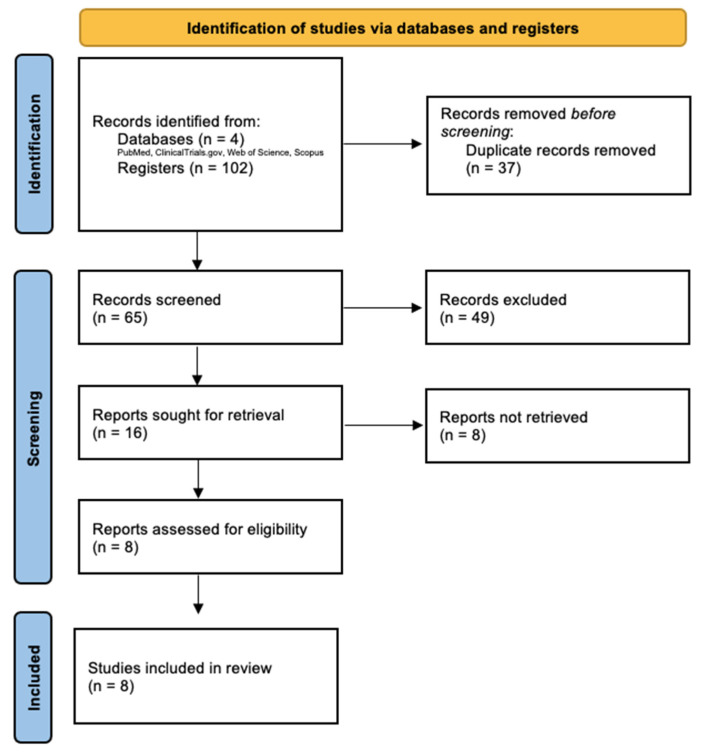
PRISMA flow-diagram of database search, study selection and included studies in this systematic review with meta-analysis.

**Figure 2 vaccines-10-01560-f002:**
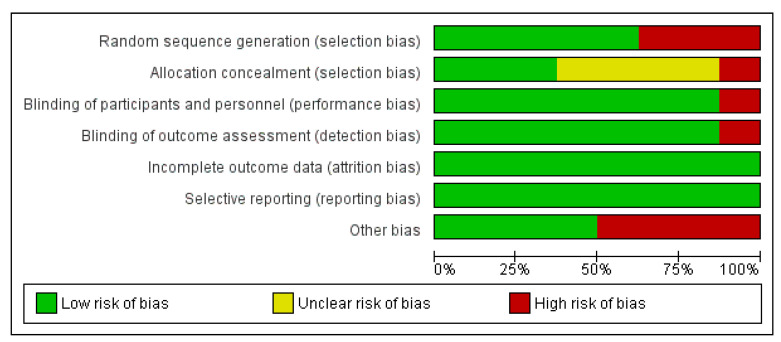
Risk of bias graph: review author’s judgments about each risk of bias item presented as percentages across all included studies.

**Figure 3 vaccines-10-01560-f003:**
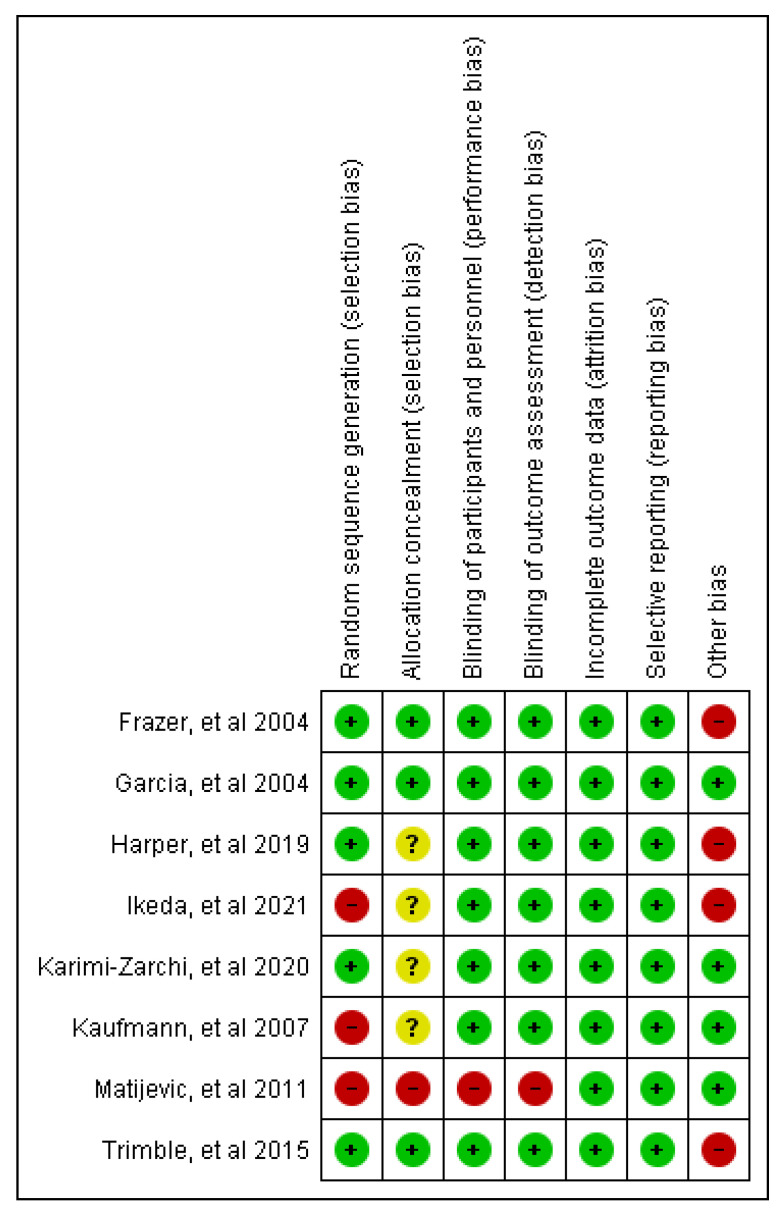
Risk of bias summary: review author’s judgments about each risk of bias item for each included study [35,36,37,38,39,40,41,42].

**Figure 4 vaccines-10-01560-f004:**
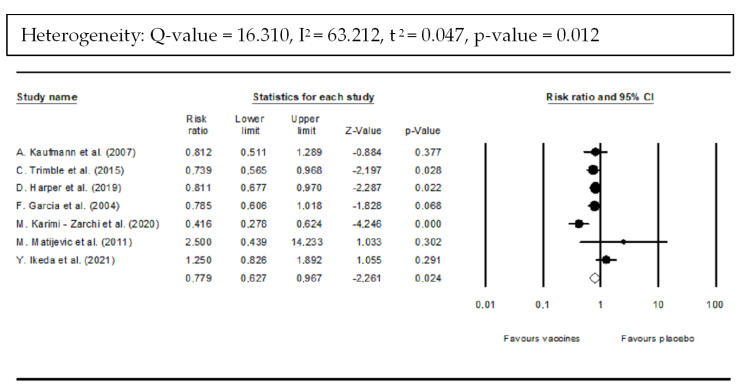
Forest plot of histopathological downgrading results after vaccination compared with placebo. Downgrading was considered an histological improvement to normal or CIN 1 [35,36,37,38,41,42].

**Figure 5 vaccines-10-01560-f005:**
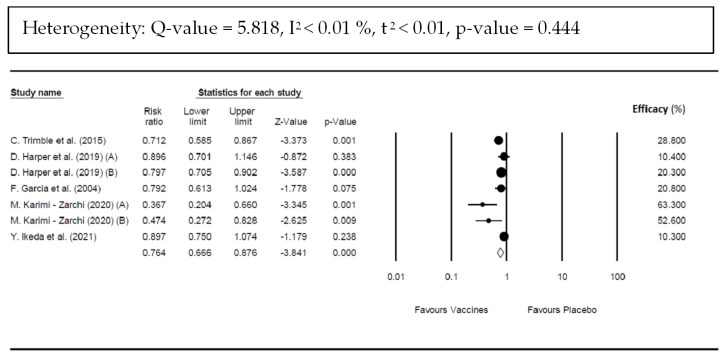
Forest plot of vaccines’ efficacy against high-grade cervical intraepithelial neoplasia (CIN 2/3) in women compared with placebo. Data were obtained by the presence of CIN 2/3 confirmed cases in vaccinated group versus unvaccinated group [35,36,37,39,41].

**Figure 6 vaccines-10-01560-f006:**
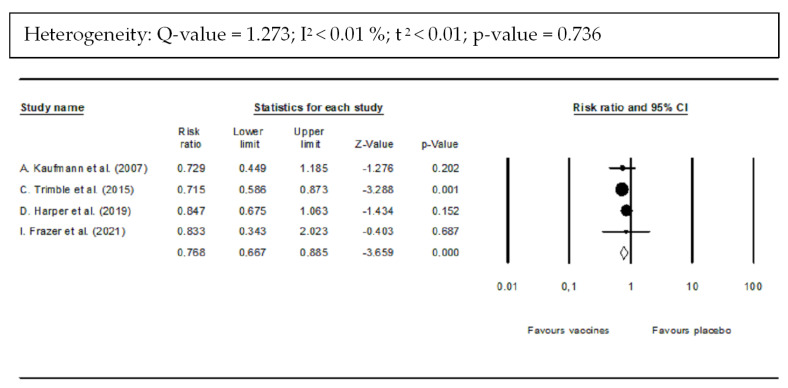
Forest plot of DNA clearance after vaccination compared with placebo. PCR amplification was utilized to assess HPV DNA in cervical biopsies [35,37,38].

**Figure 7 vaccines-10-01560-f007:**
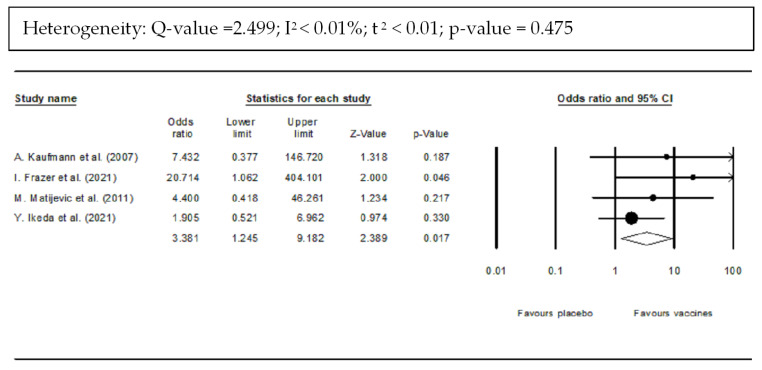
Forest plot of HPV-T cell response after vaccination compared with placebo [38,39,42].

**Figure 8 vaccines-10-01560-f008:**
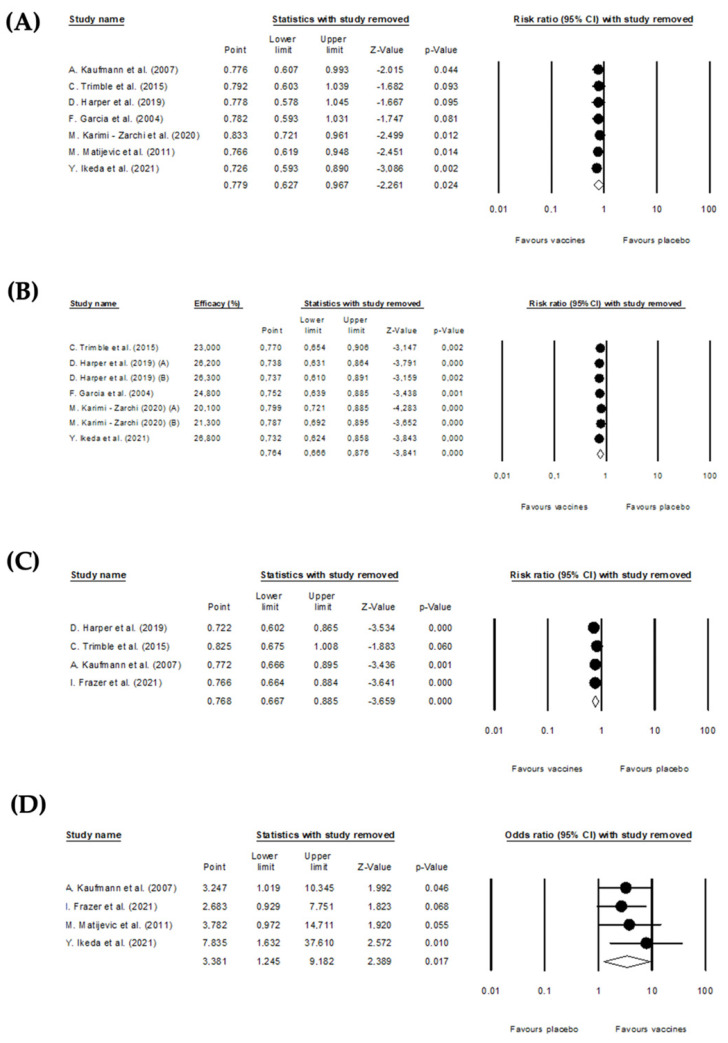
Sensitivity analysis excluding one or more studies from the analysis to see how this affected the results. (**A**) Histopathological regression to CIN ≤ 1 [35,36,37,38,41,42], (**B**) efficacy for complete resolution [35,36,37,39,41], (**C**) DNA clearance [35,37,38], (**D**) HPV-T cell response [35,38,42].

**Figure 9 vaccines-10-01560-f009:**
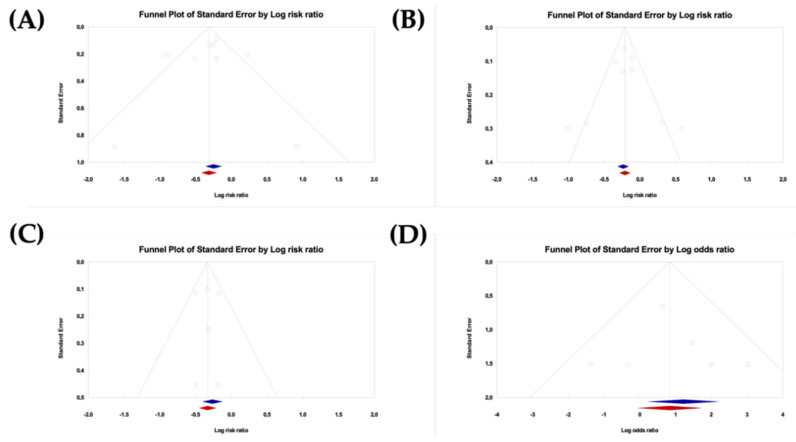
Funnel plots for the outcomes of this meta-analysis adjusted by Trim and Fill approach. (**A**) Histological improvement to normal or CIN 1, (**B**) Vaccine efficacy for complete resolution, (**C**) DNA clearance, (**D**) HPV-T cell responses after vaccination.

**Table 1 vaccines-10-01560-t001:** Description of the vaccines developed for the regression of CIN 3 selected for this systematic review with meta-analysis.

Author	Year	Vaccine			Number of Participants		CIN	Ref
		Name	Type	Administration Method	Control	Experimental		
Cornelia L. Trimble et al.	2016	VGX-3100	pDNA encoding optimized synthetic consensus E6 and E7 genes of HPV 16 and HPV 18	Intramuscular injection followed by Electroporation	40	114	2/3	[35]
Mojgan Karimi-Zarchi et al.	2020	Gardasil	Quadrivalent vaccine based on L1 VLPs from HPV 6, 11, 16, and 18	Intramuscular injection	69	83	1+	[36]
Diane M. Harper et al.	2019	Tipapkinogen Sovacivec vaccine	Viral vector expressing human cytokine IL-2 and modified HPV 16 E6 and E7	Subcutaneous injection	63	129	2/3	[37]
Andreas M. Kaufmann et al.	2007	Chimeric virus-like particle (CVLP) vaccine	CVLP of carboxy-terminally truncated HPV16L1 protein fused to the amino-terminal part of the HPV16 E7 protein	Subcutaneous injection	12	23	2/3	[38]
Ikeda, Y. et al.	2021	GLBL101c	Heat-attenuated recombinant *L. casei* expressing mutated HPV16 E7	Oral administration	19	19	2/3	[39]
Ian H. Frazer et al.	2004	-	HPV16 E6E7 fusion recombinant protein and ISCOMATRIX adjuvant	Intramuscular injection	7	24	1+	[40]
Francisco Garcia et al.	2004	ZYC101a	pDNA expressing antigenic regions of E6 and E7 of HPV 16 and 18	Poly-lactide co-glycolide (PLG) microparticles with intramuscular injection	50	111	2/3	[41]
Mark Matijevic et al.	2011	ZYC101a	pDNA expressing antigenic regions of E6 and E7 of HPV 16 and 18	Poly-lactide co-glycolide (PLG) microparticles with intramuscular injection	5	21	2/3	[42]

**Table 2 vaccines-10-01560-t002:** Adjusted values of the pooled effect sizes for the absence of publication bias (Duval and Tweedie’s Trim and Fill).

Outcomes	Number of Studies	Number of Patients	Pooled Effect Observed (95% CI)	*p*-Value	I^2^ (%)	Model Used	Studies Trimmed	Pooled Effect Adjusted (95% CI)
**Histological improvement to normal or CIN 1**	7	729	RR: 0.779 (0.697 to 0.967)	0.024	63.612	Random	3	RR: 0.692 (0.553 to 0.866)
**Vaccine Efficacy for complete resolution**	7	675	VE: 23.6% (12.4% to 33.4%)	<0.001	56.516	Random	2	VE: 18.7% (4.2% to 30.8%)
**DNA clearance**	4	386	RR: 0.768 (0.667 to 0.885)	<0.001	0	Fixed	2	RR: 0.717 (0.637 to 0.801)
**HPV-T cell response after vaccination**	4	357	OR: 3.381 (1.245 to 9.182)	0.017	0	Fixed	2	OR: 2.371 (0.857 to 6.558)

**Table 3 vaccines-10-01560-t003:** Assessment of publication bias by the Egger’s regression test.

Outcomes	Regression Intercept	95% CI	t-Value	df	*p*-Value
**Histological improvement to normal or CIN 1**	0.53	−3.38 to 4.45	0.35	5	0.74
**Vaccine Efficacy for complete resolution**	−2.21	−4.95 to 0.54	2.06	5	0.09
**DNA clearance**	0.15	−3.61 to 3.90	0.17	2	0.88
**HPV-T cell response after vaccination**	2.07	−0.23 to 4.37	3.88	2	0.06

## Data Availability

Not applicable.

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
