# Peer review of "The Effectiveness of Therapeutic Vaccines for the Treatment of Cervical Intraepithelial Neoplasia 3: A Systematic Review and Meta-Analysis"

_vaccines, 2022, doi:10.3390/vaccines10091560_

Round 1

Reviewer 1 Report

The authors performed a systematic review of the literature for RCTs of therapeutic vaccines against HPV induced CIN 3. Their search strategy yielded a total of 8 papers that encompassed their search criteria. Their analysis found that therapeutic vaccines are effective against the treatment of CIN 3. This reviewer found this study to be very interesting and covering a very topical issue in the field. I do have some issues with the manuscript in its current form, and therefore recommend major revisions prior to acceptance for publication.

Issues: 

1. In the title, you listed CIN 2/3?; however, in the abstract and in the aims of the study you explicitly mention CIN 3 only.

2. There are english spelling and grammar issues throughout the manuscript.

3. That number has changed, and there are now over 400 types of HPVs that have been identified.

4. Lines: 69 - 70: "Thus, the viral genome is encapsulated, and the virions are released to infect other cells". This makes it seem that HPV is a lytic virus. HPVs replicative cycle is not lytic. It does not break-open the cell and release viral progeny. Instead, it sloughs off with the dead epithelia that naturally shed from human skin.

5. Lines 81-81: That is not how you correctly describe vaccine valence: 

two-valent = bivalent
fourth-valent = quadrivalent
ninth-valent = nonavalent

6. Lines 83-85: There is no L2 in the commercially available vaccines you mentioned (gardasil -4, -9, and cervarix). They are just based on recombinant L1.

7. Lines 87 - 90: Where did you get those stats for the number of patients that die (50%)?. There is no reference, so please include one, because as far as I am aware 50% seems very high.

8. All figure legends and table titles need a lot more details than just a simple sentence.

9. There are two Figure 1s in the manuscript.

10. Table 2 is not a table. Those are figures.

11. Table 3 is not a table. 

Author Response

The authors would like to acknowledge the careful evaluation and pertinent Reviewer’s comments and the possibility to improve our manuscript. All the questions were answered and the recommended modifications were made, being properly highlighted in yellow in the revised manuscript file. The authors look forward to hearing from you regarding the suitability of the revised manuscript for publication in the Vaccines Journal.

Reviewer 1:

The authors performed a systematic review of the literature for RCTs of therapeutic vaccines against HPV induced CIN 3. Their search strategy yielded a total of 8 papers that encompassed their search criteria. Their analysis found that therapeutic vaccines are effective against the treatment of CIN 3. This reviewer found this study to be very interesting and covering a very topical issue in the field. I do have some issues with the manuscript in its current form, and therefore recommend major revisions prior to acceptance for publication.

  1. In the title, you listed CIN 2/3?; however, in the abstract and in the aims of the study you explicitly mention CIN 3 only.

Response: We deeply appreciate the Reviewer comment that allow us to improve the manuscript and the title was changed to CIN 3.

  1. There are english spelling and grammar issues throughout the manuscript.

Response: The English spelling and grammar was revised throughput the manuscript.

  1. That number has changed, and there are now over 400 types of HPVs that have been identified.

Response: We appreciate the Reviewer for clearing up this lapse in literature, allowing that the manuscript is in line with the most recent and updated data. The information has been changed in the manuscript along with the addition of a new reference to complement the added data (please see page 2, line 55).

  1. Moody, C.A. Regulation of the Innate Immune Response during the Human Papillomavirus Life Cycle. Viruses 2022, 14, 1797, doi:10.3390/v14081797

  1. Lines: 69 - 70: "Thus, the viral genome is encapsulated, and the virions are released to infect other cells". This makes it seem that HPV is a lytic virus. HPVs replicative cycle is not lytic. It does not break-open the cell and release viral progeny. Instead, it sloughs off with the dead epithelia that naturally shed from human skin.

Response: We deeply thank the Reviewer for helping us to make the Introduction of this manuscript as correct as possible. In this way, the final phase of the replicative cycle of HPVs was changed, according to the Reviewer’s suggestion, and supplemented with a new reference (please see page 2, line 69).

  1. Muñoz-Bello, J.O.; Carrillo-García, A.; Lizano, M. Epidemiology and Molecular Biology of HPV Variants in Cervical Cancer: The State of the Art in Mexico. Int J Mol Sci 2022, 23, 8566, doi:10.3390/ijms23158566.

  1. Lines 81-81: That is not how you correctly describe vaccine valence: 

Response: We would like to thank the Reviewer for clarifying the correct way to describe vaccine valence. The manuscript has been corrected, considering the Reviewer's suggestion (please see page 2, line 80).

  1. Lines 83-85: There is no L2 in the commercially available vaccines you mentioned (gardasil -4, -9, and cervarix). They are just based on recombinant L1.

Response: We sincerely apologize to the Reviewer for this misunderstanding. In fact, the commercially available vaccines are based on recombinant L1 protein and this information was corrected (please see page 2, line 84).

Lines 87 - 90: Where did you get those stats for the number of patients that die (50%)?. There is no reference, so please include one, because as far as I am aware 50% seems very high.

Response: We deeply thank the Reviewer for the possibility to improve our manuscript and we sincerely apologize for this misunderstanding. The manuscript was corrected (please see page 2, line 89.

  1. All figure legends and table titles need a lot more details than just a simple sentence.

Response: We deeply thank the Reviewer for the possibility to improve our manuscript, the figure legends and table titles were improved as suggested by the Reviewer.

  1. There are two Figure 1s in the manuscript.

Response: At this moment, we have only one figure 1 corresponding to the PRISMA flow-diagram of database search, study selection and included studies in this meta-analysis, please see page 5.

  1. Table 2 is not a table. Those are figures.
  2. Table 3 is not a table. 

Responses: We are grateful to the Reviewer for these comments. Tables 2 and 3 were converted and formatted as figures.

Reviewer 2 Report

Dera Editor,

the manuscript by  Ventura et al., are will written and my provide a new information for the reader of the journal. The data are presented in best form and the editting is very good. Accordingly, the manuscript can be published in the present form.

Many thanks

Author Response

The manuscript by Ventura et al., are will written and my provide a new information for the reader of the journal. The data are presented in best form and the editting is very good. Accordingly, the manuscript can be published in the present form.

Many thanks.

Response: The authors would like to express our deep appreciation for the comment made by the Reviewer about the submitted manuscript, and we agree that new information will be forwarded on the subject of therapeutic vaccines. We also appreciate the kindness with which the Reviewer complimented all the structure and work undertaken to obtain this manuscript.

Reviewer 3 Report

Comments to Authors 

            This study showed that therapeutic vaccines are efficient in the treatment of CIN 3, even after accounting for publication bias.

           Authors are kindly requested to emphasize the current concepts about these issues in the context of recent knowledge and the available literature. This articles should be quoted in the References list.

References

11.      The Efficacy of Therapeutic DNA Vaccines Expressing the Human Papillomavirus E6 and E7 Oncoproteins for Treatment of Cervical Cancer: Systematic Review. Vaccines (Basel). 2021; 10 (1): 53. Published 2021 Dec 31. doi:10.3390/vaccines10010053.

22.      Therapeutic DNA Vaccines against HPV-Related Malignancies: Promising Leads from Clinical Trials. Viruses. 2022;14(2):239. Published 2022 Jan 25. doi:10.3390/v14020239.

33.      HPV Vaccination in Women with Cervical Intraepithelial Neoplasia Undergoing Excisional Treatment: Insights into Unsolved Questions. Vaccines (Basel). 2022;10(6):887. Published 2022 Jun 1. doi:10.3390/vaccines10060887.

44.      Role of human papillomavirus (HPV) vaccination on HPV infection and recurrence of HPV related disease after local surgical treatment: systematic review and meta-analysis. BMJ. 2022;378:e070135. Published 2022 Aug 3. doi:10.1136/bmj-2022-070135.

Author Response

This study showed that therapeutic vaccines are efficient in the treatment of CIN 3, even after accounting for publication bias.

Authors are kindly requested to emphasize the current concepts about these issues in the context of recent knowledge and the available literature. This articles should be quoted in the References list.

References

  1. The Efficacy of Therapeutic DNA Vaccines Expressing the Human Papillomavirus E6 and E7 Oncoproteins for Treatment of Cervical Cancer: Systematic Review. Vaccines (Basel). 2021; 10 (1): 53. Published 2021 Dec 31. doi:10.3390/vaccines10010053.
  2. Therapeutic DNA Vaccines against HPV-Related Malignancies: Promising Leads from Clinical Trials. Viruses. 2022;14(2):239. Published 2022 Jan 25. doi:10.3390/v14020239.
  3. HPV Vaccination in Women with Cervical Intraepithelial Neoplasia Undergoing Excisional Treatment: Insights into Unsolved Questions. Vaccines (Basel). 2022;10(6):887. Published 2022 Jun 1. doi:10.3390/vaccines10060887.
  4. Role of human papillomavirus (HPV) vaccination on HPV infection and recurrence of HPV related disease after local surgical treatment: systematic review and meta-analysis. BMJ. 2022;378:e070135. Published 2022 Aug 3. doi:10.1136/bmj-2022-070135.

Response: We deeply thank the Reviewer for the possibility to improve the discussion of our manuscript including these references, please see page 16, line 433.

Round 2

Reviewer 1 Report

The authors have adequately addressed all my comments and concerns. The manuscript is now suitable for publication. Thank you!